# DemoNSF: A Multi-task Demonstration-based Generative Framework for Noisy Slot Filling Task

**Guanting Dong**[1*], **Tingfeng Hui**[1*], **Zhuoma GongQue**[1], **Jinxu Zhao**[1], **Daichi Guo**[1],
**Gang Zhao**[1], **Keqing He**[2], **Weiran Xu**[1*]

[1]Beijing University of Posts and Telecommunications, Beijing, China
[2]Meituan, Beijing, China
{dongguanting,huitingfeng,xuweiran}@bupt.edu.cn

## Abstract

Recently, prompt-based generative frameworks have shown impressive capabilities in sequence labeling tasks. However, in practical dialogue scenarios, relying solely on simplistic templates and traditional corpora presents a challenge for these methods in generalizing to unknown input perturbations. To address this gap, we propose a multi-task demonstration-based generative framework for noisy slot filling, named **DemoNSF**. Specifically, we introduce three noisy auxiliary tasks, namely noisy recovery (NR), random mask (RM), and hybrid discrimination (HD), to implicitly capture semantic structural information of input perturbations at different granularities. In the downstream main task, we design a noisy demonstration construction strategy for the generative framework, which explicitly incorporates task-specific information and perturbed distribution during training and inference. Experiments on two benchmarks demonstrate that DemoNSF outperforms all baseline methods and achieves strong generalization. Further analysis provides empirical guidance for the practical application of generative frameworks. Our code is released at https://github.com/dongguanting/Demo-NSF.

## 1 Introduction

The slot filling (SF) task in the goal-oriented dialog system aims to identify task-related slot types in certain domains for understanding user utterances. Recently, traditional discriminative and generative models (Liu and Lane, 2015, 2016; Goo et al., 2018; Niu et al., 2019; He et al., 2020a; Yan et al., 2021a; Wang et al., 2022b; Hao et al., 2023) have shown remarkable ability in slot filling. Despite their powerful capabilities, the high performance of these models heavily depends on the consistency of data distribution between the training and test sets.

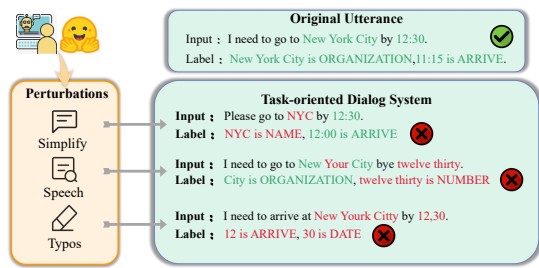

Figure 1: The impact of diverse input perturbations on the slot filling system in real scenarios.

When faced with the uncertainty and diversity of human language expression (Wu et al., 2021), these perturbations significantly impact the SF model's generalization ability, thereby hindering its application in practical dialogue scenarios.

In real dialogue systems, models often encounter a wide range of input perturbations and errors made by humans. As illustrated in Figure 1, users may interact with the dialogue system in ways that deviate from the standard input format and even simplify their queries to convey the same intent, all due to diverse human expression habits. Furthermore, errors originating from the upstream input system may introduce disturbances to the downstream model (e.g. Typos from keyboard input, Speech errors from ASR systems). Existing slot filling models are typically pre-trained and fine-tuned on perturbation-free datasets, leading to decreased performance when confronted with such situations.

Recently, existing studies (Wu et al., 2021; Moradi and Samwald, 2021a; Gui et al., 2021) have explored the issue of robustness. However, these methods are mainly designed for particular perturbations, limiting generalization ability to unknown perturbations. To capture the noisy semantic structure, PSSAT and CMDA (Dong et al., 2022a; Guo et al., 2023) further introduce additional corpus and generative models. Nevertheless, this approach carries the risk of introducing extra noise and increasing computing resource consumption. While Large language models (Brown et al., 2020; Tou-

---

*The first two authors contribute equally. Weiran Xu is the corresponding author.

vron et al., 2023; OpenAI, 2023) and prompt-based methods (Lu et al., 2022; Xie et al., 2022b) have achieved strong performance on information extraction, the exploration of these generative frameworks on diverse input perturbations remains a blank area, hindering their application in realistic task-oriented dialogue systems.

To address this limitation, we propose a multi-task demonstration-based generative framework for noisy slot filling tasks, named **DemoNSF**. Specifically, we design three noisy auxiliary tasks, namely noisy recovery (NR), random mask (RM), and hybrid discrimination (HD), to boost the performance against input perturbations in different levels. NR aims to capture the mapping relationship between fine-grained noisy and clean data. RM implicitly learns the slot entity distribution of perturbed data during the process of mask infilling. HD assists generative models consider global information while implicitly capturing the semantic characteristics unique to perturbed data. In the downstream process, we formulate the SF task as a sequence-to-sequence generation guided by noisy task demonstrations. In detail, DemoNSF selects a semantically similar example for each query from a noisy candidate pool, converts it into a natural demonstration sentence, and encodes the demonstration along with the input text by integrating noisy semantic information. With the boost of noisy auxiliary tasks and demonstrations, DemoNSF learns the semantic structure of perturbations from both explicit and implicit levels. Our contributions are three-fold:

1) To the best of our knowledge, we are the first to comprehensively investigate the effects of diverse input perturbations on generative frameworks in slot filling tasks and further validate the vulnerability of existing prompt-based generative methods when confronted with different human expressions.

2) We propose a simple but unified multi-task demonstration-based generative framework, which includes three novel noisy auxiliary tasks and a noisy demonstration construction strategy, to enhance the model's robustness and adaptability to perturbed inputs in real-world dialogue scenarios.

3) Experiments on two benchmarks demonstrate that our method outperforms all baseline methods and achieves strong generalization. The extensive analysis also provides empirical guidance for the practical application of generative frameworks.

## 2 Method

In this section, we introduce the overall framework of our proposed DemoNSF. We first briefly describe the problem definition against input perturbations in the slot filling task. Next, we propose three distinctive noisy auxiliary tasks for diverse perturbations. Finally, we present a novel noisy demonstration construction strategy. We will introduce these in the following subsections[1].

### 2.1 Problem Definition

Given an input utterance $X = \{x_1, x_2, \ldots, x_N\}$ and its corresponding slot type set $S = \{s_1, ..., s_m\}$, the slot filling task aims to extract all the entities in $X$. For the noisy slot filling task, we formulate the input perturbation process in the real scenario as $[(X', S') = \mathcal{P}(X, S)]$, The model's robustness is evaluated on the perturbed test dataset $\{(X', Y')\}$ but with no access to the input perturbation process $\mathcal{P}(\cdot)$ or perturbed data during the training phase. In this paper, We use $D_{clean}$, $D_{aug}$, and $D_{test}$ to denote clean data, augmented data, and test data.

### 2.2 Multi-level Data Augmentation

Figure 2 demonstrates how we construct our noisy candidate pool using NLPAug (Ma, 2019), enabling the input utterance of the clean training set into an augmented dataset comprising three distinct levels: **character-level**, **word-level**, and **sentence-level** augmentation. Specifically, at the character level, we incorporate random operations such as character addition, deletion, and substitution within a token, governed by a probability parameter denoted as $p$. Moving to the word level, we introduce random word deletion, insertion, and replacement, along with the substitution of words with homophones within a sentence, again governed by the probability parameter $p$[2]. Furthermore, at the sentence level, we substitute sentences with synonymous alternatives.

### 2.3 Noisy Auxiliary Tasks

The performance of the noisy slot filling task highly depends on the prior knowledge of the distribution of input perturbations. In this section, we introduce three novel noisy auxiliary tasks:

**Noisy Recovery (NR).** Given a character-level augmented utterance $X_{char}^{aug} = \{x_1, x_2, ..., x_m^{aug},$

---

[1]Training, and Inference can be found in Appendix

[2]$p$ is an empirical parameter, we set it to 0.3

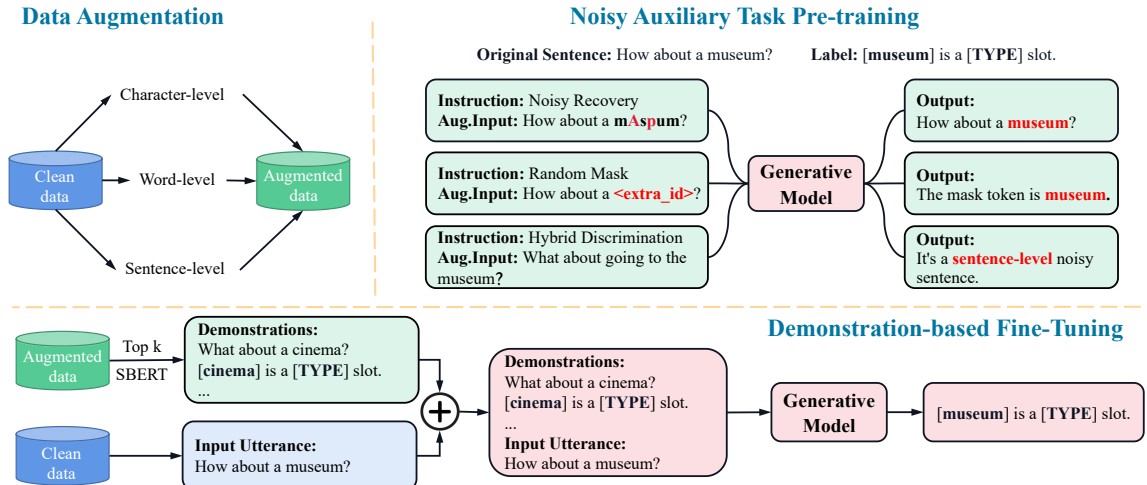

Figure 2: The overall architecture of our proposed approach DemoNSF.

..., $x_N$}, where $x_m^{aug}$ represents the augmented token with character-level augmentation, the objective of the NR task, as illustrated in Figure 2, is to restore $X_{char}^{aug}$ to its corresponding clean utterance $X$. This task enables the model to capture the mapping relationship between fine-grained input perturbations and their clean equivalents, thereby enhancing the model's capability to represent fine-grained noisy data. Hence, the loss function can be formulated as:

$$L_{NR} = \frac{1}{B} \sum_{j=1}^{B} \sum_{i=1}^{N} CE(X_{ji}, X_{char,ji}^{aug}) \quad (1)$$

where B and N denote the batch size and sequence length, respectively.

**Random Mask (RM).** Inspired by the concept of masked language modeling (MLM) introduced in BERT (Devlin et al., 2019), we present our random mask-filling task. Specifically, given an utterance in $D_{aug}$, we randomly mask one entity with the special *[MASK]* symbol, resulting in $X_{mask}^{aug} = \{x_1^{aug}, ..., [MASK], ..., x_N^{aug}\}$. We aim to restore the *[MASK]* token to its original value. The mask-filling procedure enables the model to implicitly incorporate the semantic distribution of slot entities within the perturbed data. Hence, the loss function of the RM task can be defined as:

$$L_{RM} = \frac{1}{B} \sum_{j=1}^{B} CE(y_m, P([MASK])) \quad (2)$$

where $y_m$ denotes the original token and *[MASK]* represents the logits of *[MASK]* token.

**Hybrid Discrimination (HD).** To further address coarse-grained input perturbations, we propose the HD task. In detail, we combine $D_{clean}$ and $D_{aug}$ to create a mixed dataset, denoted as $D_{mix}$. We randomly select utterances from $D_{mix}$ and assign distinct labels based on whether the chosen utterance is clean or has different levels of perturbation. As shown in Figure 2, the generative model can implicitly capture the unique semantic distribution of perturbed data while considering global information by discriminating between inputs with and without perturbation. The loss function $L_{HD}$ is the same as $L_{NR}$.

Therefore, the overall loss function $L$ is defined as:

$$L = \alpha L_{NR} + \beta L_{RM} + \gamma L_{HD} \quad (3)$$

where $\alpha$, $\beta$, and $\gamma$ represent the weights of NR, RM, and HD task loss functions, respectively.

### 2.4 Noisy Demonstration Construction

Different from prior demonstration-based work (Min et al., 2022), we select examples $s$ from $D_{aug}$ instead of $D_{clean}$ for each input $X$ to incorporate perturbed semantic information into the model. For retrieval, we employ SBERT (Reimers and Gurevych, 2019) which independently produces $[CLS]$ embeddings for both $X$ and $s$, and compute their similarity scores to rank $s$. Subsequently, we select the top-$k$ examples to construct the noisy demonstrations $\hat{X}$ and concatenate them with the input $X$ to form the complete input $[\hat{X}; X]$. Our demonstration template is shown below:

"**Demonstrations:** [Retrieved Noisy Utterances]. [Text Span] is [Slot Type]. **Input Utterance:** [Original Input]."

| Methods | Clean | Sentence-level | | | Character-level | Word-level | Perturbed-Avg. |
|---|---|---|---|---|---|---|---|
| | | Verbose | Paraphrase | Simplification | Typos | Speech | |
| GPT2 | 95.37 | 80.52 | 85.66 | 82.98 | 60.19 | 77.78 | 77.43 |
| BART | 95.28 | 77.87 | 82.72 | 82.95 | 53.90 | 73.57 | 74.20 |
| T5 | 95.49 | 81.34 | 89.13 | 83.73 | 62.43 | 81.13 | 79.55 |
| BARTNER | 94.88 | 78.00 | 88.55 | 85.04 | 65.37 | 72.65 | 77.93 |
| LightNER | 95.30 | 78.85 | 87.65 | 84.90 | 57.68 | 71.61 | 76.14 |
| InstructionNER | 95.67 | 81.57 | 88.45 | 85.29 | 65.34 | 80.13 | 80.16 |
| DemoNSF(GPT2) | 95.66(±0.4) | 81.95(±0.3) | 87.63(±1.1) | 87.02(±0.2) | 69.75(±0.3) | 86.31(±0.7) | 82.53(±0.5) |
| DemoNSF(BART) | 95.71(±1.3) | 78.83(±0.7) | 88.29(±0.8) | 86.01(±0.7) | 65.60(±0.3) | 82.48(±0.4) | 80.24(±1.1) |
| DemoNSF(T5) | **95.72**(±0.5) | **82.37**(±1.2) | **89.98**(±1.1) | **89.49**(±0.7) | **76.63**(±0.5) | **87.55**(±0.7) | **85.20**(±0.9) |

Table 1: F1 scores with standard deviations under 5 different input perturbations on RADDLE.

| Methods | Char+Word | Char+Sen | Word+Sent | Char+Word+Sen |
|---|---|---|---|---|
| | Ent.+Sub. | Ent.+App. | App.+Sub. | Ent.+Sub.+App. |
| BART | 58.00 | 47.27 | 50.28 | 38.36 |
| T5 | 57.44 | 56.79 | 73.47 | 47.93 |
| BARTNER | 54.83 | 49.10 | 58.92 | 42.25 |
| LightNER | 42.34 | 35.82 | 45.44 | 27.00 |
| InstructionNER | 57.87 | 58.89 | 74.45 | 50.75 |
| Ours(BART) | 61.27(±0.5) | 51.26(±0.9) | 68.72(±0.5) | 44.27(±0.9) |
| Ours(T5) | **63.59**(±0.3) | **63.94**(±1.2) | **77.69**(±0.7) | **55.12**(±0.3) |

Table 2: F1 scores with standard deviations under 4 kinds of mixed perturbations on SNIPS.

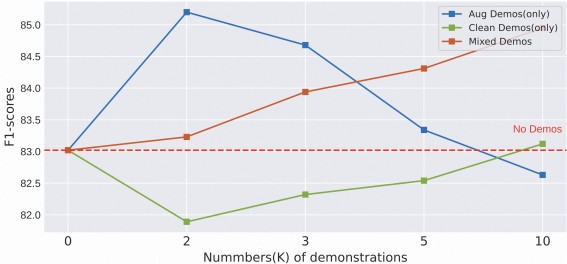

Figure 3: Performance comparison of different types of demonstrations

## 3 Experiment

### 3.1 Datasets

Based on RADDLE (Peng et al., 2021) and SNIPS (Coucke et al., 2018), we adopt the evaluation set provided by Dong et al., which includes two different perturbation settings. For single perturbations setting, we include five types of noisy utterances (character-level: **Typos**, word-level: **Speech**, and sentence-level: **Simplification**, **Verbose**, and **Paraphrase**) from RADDLE. For mixed perturbations setting, we utilize TextFlint (Gui et al., 2021) to introduce character-level perturbation (**EntTypos**), word-level perturbation (**Subword**), and sentence-level perturbation (**AppendIrr**) and combine them to get a mixed perturbations dataset[3].

### 3.2 Main Results.

Table 1 shows the main results of DemoNSF and comparison baselines under a single perturbation setting. We make the following observations:

(1) When faced with perturbations of different granularities, generative models suffer severe performance degradation, especially in Typos (GPT2: 35.18% | BART: 41.38% | T5: 33.06%) and Speech (GPT2: 17.57% | BART: 21.71% | T5: 14.36%), which indicates that generative models have poor robustness against fine-grained perturbations.

(2) DemoNSF(T5) shows remarkable superiority under different kinds of input perturbations while maintaining the best performance on the clean test data. For the fine-grained perturbations , our method achieves a significant improvement of 11.29% and 7.42% in Typos and Speech compared with InstructionNER. DemonNSF maintains strong performance for coarse-grained perturbations, especially with a 4.2% improvement in Simplification. These results clearly demonstrate that DemonNSF effectively captures the mapping relationship between fine-grained noisy and clean data while also considering the ability to generalize to coarse-grained global semantic perturbations.

(3) DemonNSF is a plug-and-play method that can achieve good results on different backbones. Specifically, we replace the backbone with BART/GPT and also get the good performances compared with the corresponding baseline. The results of the backbone ablation further demonstrate our approach remarkably enhances the robustness of generative models when facing perturbations.

### 3.3 Mixed Perturbations Scenario.

In real dialogue scenarios, mixed perturbations often appear in one utterance simultaneously. To further verify the effectiveness of DemoNSF in more realistic scenarios, we conduct the mixed perturbations experiment. As shown in Table 2, DemoNSF significantly outperforms other baselines in all

---

[3]Due to space limitations, detailed experimental settings (Baselines, Datasets..) can be found in the Appendix B.1.

| Methods | Clean | Sentence-level | | | Character-level | Word-level | Perturbed-Avg. |
|---|---|---|---|---|---|---|---|
| | | Verbose | Paraphrase | Simplification | Typos | Speech | |
| Text-davinci-003 | 43.09 | 34.26 | 39.34 | 38.42 | 40.12 | 37.18 | 38.54 |
| ChatGPT | 71.43 | 40.65 | 60.00 | 55.56 | 65.54 | 55.56 | 57.21 |
| ChatGPT + Clean Demos | 71.31 | 61.01 | 57.81 | 53.43 | 65.03 | 61.71 | 62.32 |
| ChatGPT + Aug Demos | 68.21 | **65.04** | **70.56** | **58.82** | 73.03 | **63.77** | **68.34** |
| ChatGPT + Mixed Demos | **76.92** | 58.73 | 68.26 | 58.61 | **74.19** | 57.78 | 65.36 |

Table 3: The evaluation on ChatGPT under 5 different input perturbations on Raddle.

two-level perturbations, especially achieving over 63% F1-score in fine-grained mixed perturbations. Even with the joint interference of 3 perturbations, DemoNSF can still maintain a 4.37% improvement compared with baseline, which further validates the stability of DemoNSF in challenging settings.

### 3.4 Impact of Different Demonstrations.

Figure 3 shows the impact of the number of different types of demonstrations under single perturbations. We have the following findings: (1) DemoNSF exhibits a significant performance gain with only two augmented samples while its performance severely decreases as the number increases. This may be because diverse augmented instances can help the model explicitly fuse noisy semantic distribution (Xie et al., 2022a) while the sample diversity exceeding a certain threshold may even bring additional noise. (2) Clean demonstrations only bring slightly improves as the number increases, which indicates that clean samples only provide some task general information(e.g. entity distributions, slot-value mapping) for prompting. (3) Retrieved demonstrations from the mixed data pool show a stable performance gain, which further confirms the mutual promotion between noisy semantic distribution and task general information, and provides guidance for the robustness of prompt-based generative models.

### 3.5 The ICL Evaluation on ChatGPT.

In order to further validate the effectiveness of our noisy demonstration strategy on the large-scale generative framework, we conduct experiments on ChatGPT and Text-davinci-003 (Brown et al., 2020). We directly use them to do inference based on in-context learning (ICL) (Dong et al., 2022b; Brown et al., 2020; Min et al., 2022) on RADDLE, which means language models make predictions only based on the conditioned demonstration examples without any training samples.

Table 3 illustrates the overall results of Chat-GPT under 5 different single perturbations. We draw the following findings: (1) ChatGPT and Text-davinci-003 perform poorly on diverse input perturbations, which far behind the finetune SOTA methods (DemoNSF, Instruction NER) presented in Table 1. The possible reason is that large language models are usually pre-trained on the large-scale general training corpus, making it difficult to adapt well to specific domain perturbation data in a zero-shot setting. (2) Compared with baselines and traditional clean demonstration retrieval methods, selecting instances from both the augmented and mixed demonstration candidate pools can significantly improve the overall performance. This finding is consistent with our conclusion in the section 3.4, proving the effectiveness of incorporating noisy semantic structures in addressing input perturbations. (3) From the perspective of different perturbations, both two types of noisy demonstration strategies show significant improvements in fine-grained perturbations (over 8% improvement in Typos). However, the improvement is not obvious in coarse-grained perturbations, especially in speech errors. This phenomenon indicates that noisy demonstrations are more suitable for fitting the distribution of fine-grained perturbations, while there is still much room for improvement in coarse-grained perturbations that severely disrupt the contextual semantics and slot mentions of the original input. This finding poses further challenges for exploring the robustness of large language models, which will also be the focus of our future research.

## 4 Conclusion

In this paper, we propose a unified multi-task demonstration-based generative framework for noisy slot filling tasks. Specifically, we introduce three novel noisy auxiliary tasks and a noisy demonstration construction strategy for the generative framework, which aims to learn the semantic structure of perturbations from both explicit and implicit levels. Experiments on two benchmarks show the effectiveness of DemoNSF, Further analysis provides empirical guidance for the practical application of the generative framework.

## Limitations

In order to capture semantic structural information of input perturbations at different granularities, we introduce three novel noisy auxiliary tasks in the pre-training stage, which may consume more GPU memory than traditional methods. This drives us to further improve the overall memory efficiency of the framework. Also, our method mainly focuses on the slot filling task. However, we believe it is possible to extend our work to other scenarios, such as few-shot settings, and zero-shot settings. We also reserve them for our future research.

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

## A  Method Details

### A.1  Training and Inference

For the multi-level data augmentation, we utilize NLPAug (Ma, 2019) to construct a noisy candidate data pool from the clean data pool at different levels. During the upstream pre-training stage, we adopt a multi-task training strategy, where the overall loss function is denoted as (13).

In the downstream demonstration-based fine-tuning stage, we directly feed the demonstration-based input $[\hat{X}; X]$ into the model for prediction. During inference, we also employ SBERT (Reimers and Gurevych, 2019) to retrieve task demonstrations for test data from $D_{aug}$, ensuring consistency between the training and inference stages.

## B  Experiment Details

### B.1  More Details of Datasets

Based on RADDLE (Peng et al., 2020) and SNIPS (Coucke et al., 2018), we adhere to the evaluation set provided by PSSAT (Dong et al., 2022a), which includes two settings: single perturbation and mixed perturbation.

For a single perturbation setting, RADDLE serves as a crowd-sourced diagnostic evaluation dataset that covers a wide range of real-world noisy texts for dialog systems. PSSAT(Dong et al., 2022a) extracts each type of noisy utterance (Typos, Speech, Simplification, Verbose, and Paraphrase) from RADDLE to build the test data pool. Specifically, **Typos** occur due to non-standard abbreviations or keyboard errors, while **Speech** arises from recognition and synthesis errors produced by ASR systems. **Simplification** refers to users expressing their intentions using concise words, while **Verbose** represents users employing redundant words to convey the same intention. **Paraphrase** is also prevalent among users who use different words or rephrase the text based on their language habits.

For the multi-perturbations setting, we utilize TextFlint (Gui et al., 2021) toolkit to introduce character-level noise (**EntTypos**), word-level noise (**Subword**), and sentence-level noise (**AppendIrr**). We then combine different types of noisy data to construct a multi-perturbations evaluation set. The detailed introduction of these perturbations can be found in the GitHub repository of TextFlint [4].

---

[4] https://www.textflint.io/textflint

| Methods | Clean | Sentence-level | | | Character-level | Word-level | Perturbed-Avg. |
|---|---|---|---|---|---|---|---|
| | | Verbose | Paraphrase | Simplification | Typos | Speech | |
| DemoNSF(Backbone) | 95.49 | 81.34 | 89.13 | 83.73 | 62.43 | 81.13 | 79.55 |
| $+NR$ | 95.17 | 81.13 | 88.96 | 87.98 | 67.22 | 82.61 | 81.58 |
| $+RM$ | 95.30 | 81.04 | 89.88 | 86.78 | 62.31 | 85.84 | 81.17 |
| $+HD$ | 95.39 | 81.37 | 90.36 | 87.20 | 64.61 | 82.66 | 81.24 |
| $+MT$ | 95.82 | 80.79 | 89.28 | 86.89 | 73.74 | 84.40 | 83.02 |
| $+CleanDemos$ | 95.37 | 80.28 | 88.91 | 85.94 | 69.00 | 84.62 | 81.75 |
| $+MixDemos$ | 95.90 | 80.04 | 90.71 | 87.19 | 71.23 | 82.60 | 82.35 |
| $+NoisyDemos$ | 95.08 | 80.04 | 90.34 | 85.39 | 76.41 | 85.14 | 83.46 |
| DemoNSF(Full) | 95.72 | 82.37 | 89.98 | 89.49 | 76.63 | 87.55 | 85.20 |

Table 4: The ablation study results (average F1 score%) on RADDLE. "+" denotes the backbone of DemoNSF with specific module.

## B.2 Baselines

In this paper, we focus on comparing DemoNSF with multiple state-of-the-art baselines that use a generative framework, as shown below:

**GPT2** (Radford et al., 2019) is a decoder-only framework model developed by OpenAI [5]. It is designed to generate human-like text by predicting the next word in a given sequence. GPT-2 has gained popularity for its impressive ability to generate coherent and contextually relevant text across various domains and has been used for tasks like text completion, translation, and creative writing.

**BART** is a sequence-to-sequence model architecture introduced by Lewis et al. (2019). It combines both autoregressive and denoising objectives during training to learn robust representations of input sequences.

**T5** is a pre-training model proposed by Raffel et al. (2020). It utilizes the transformer architecture and transforms various natural language processing tasks into text-to-text transfer tasks.

**BARTNER** (Yan et al., 2021b) is a pointer-based sequence-to-sequence architecture designed for NER tasks. It converts NER subtasks into a unified sequence generation task by predicting entities and their corresponding type indexes in the input sentence.

**LightNER** (Chen et al., 2022) is a pointer-based sequence-to-sequence model which builds upon the BARTNER. It introduces a prompt tuning technique that incorporates additional parameters into the attention mechanism.

**InstructionNER** (Wang et al., 2022b) is a multi-task instruction-based generative framework specifically designed for addressing few-shot NER tasks. It redefines the NER task as a natural language

---

[5]https://openai.com/

generation problem and introduces descriptive instructions and an optional mechanism to enable the model to understand different tasks and constrain the output space.

## B.3 Implementation Details

In the upstream pre-training stage, we set the batch size to 32, and our pre-training process typically takes around 1 hour for 5 epochs. In this paper, we conduct all the experiments without any hyperparameter search. For the multi-task training strategy, we assign equal weights to three noisy auxiliary tasks, i.e., set $\alpha$, $\beta$, and $\gamma$ to $\frac{1}{3}$. The corresponding learning rates are set to 1e-5. For the demonstration-based fine-tuning stage, we also set the batch size to 32 and the training takes an average of 2 hours for 5 epochs, while the learning rates are set to 5e-5. For the selection of demonstrations, we recall the top 2 instances with the highest similarity score from the noisy candidate pool.

In all experiments, we train and test various methods using NVIDIA RTX A6000 GPU. To select the best model, we evaluate the performance on the validation set using the F1 metric every 400 training steps. Experiments in Table 2 and Table 1 use base-version of T5 and BART, while we also adopt the large-version model in Table 5 on single perturbation setting. We retrieve $CleanDemos$ from $D_{clean}$, $NoiseDemos$ from $D_{aug}$ and $MixDemos$ from $D_{mix}$ for the ablation study and the experiments on investigating the impact of the number of demonstrations. We will release our code after a blind review.

| Methods | Clean | Sentence-level | | | Character-level | Word-level | Perturbed-Avg. |
|---|---|---|---|---|---|---|---|
| | | Verbose | Paraphrase | Simplification | Typos | Speech | |
| BART | 95.21 | 79.23 | 87.20 | 83.81 | 57.79 | 75.65 | 76.74 |
| T5 | 95.58 | 82.12 | 88.36 | 85.98 | 68.25 | 81.31 | 81.20 |
| BARTNER | 95.30 | 79.96 | 90.44 | 87.24 | 75.32 | 76.22 | 81.84 |
| LightNER | 96.02 | 80.32 | 90.40 | 87.97 | 67.28 | 75.57 | 80.31 |
| InstructionNER | 95.05 | 82.01 | 87.82 | 85.25 | 69.75 | 80.09 | 80.98 |
| DemoNSF(BART) | 95.77($\pm$1.4) | 81.25($\pm$0.7) | 90.56($\pm$0.1) | 88.12($\pm$0.5) | 74.20($\pm$0.7) | 84.58($\pm$0.3) | 83.74($\pm$0.4) |
| DemoNSF(T5) | **95.81($\pm$0.7)** | **83.77($\pm$0.4)** | **91.58($\pm$0.6)** | **89.78($\pm$0.2)** | **77.70($\pm$1.3)** | **87.96($\pm$0.9)** | **86.16($\pm$0.5)** |

Table 5: F1 scores with standard deviations under 5 different input perturbations on RADDLE. All the models are in large versions

## C  More Detailed Experiments

### C.1  Ablation Study

We conduct an ablation study to investigate the characteristics of the main components in DemoNSF. As shown in Table 4, we have the following observations: 1) The performance of the model improves when adding any component, which demonstrates that every part of our design is necessary. 2) For the three different granularities of perturbations, we observe significant improvements in auxiliary tasks specifically designed for each. Specifically, the NR task learns the mapping relationship between character-level perturbations and clean data, resulting in a 4.79% improvement in Typos. While the RM task implicitly captures the semantic information of slot entities during the mask-filling procedure. It achieved about 4.71% improvement under word-level perturbations (Speech). As for the HD task, it is able to capture the unique distribution information of perturbed data and significantly improves the performance of the model under coarse-grained perturbations while maintaining generalization, especially in Simplification (3.47%). 3) Adopting joint pre-training tasks ($_{+MT}$) results in a noticeable improvement compared with adding one of them, which indicates that jointly pre-training objectives have a mutually reinforcing effect (obtain 3.47% improvement on average of perturbed data). 4) We explore the ablation study of three demonstration retrieval strategies. $CleanDemos$, $MixDemos$, and $NoisyDemos$ represent retrieve demonstrations from $D_{clean}$, $D_{mix}$ and $D_{aug}$, respectively. As for $MixDemos$, We make sure to include both clean and noisy demonstrations. We find that concatenating demonstrations does yield exciting results on perturbed test data. Specifically, while $MixDemos$ is able to absorb more diverse data distributions and performs well on both clean and perturbed data, the $NoisyDemos$ used in this paper focuses on introducing the distribution information of the perturbed data, so that the generative model can learn the perturbed sentence and slot entity distribution information to the maximum extent and make it more robust.

### C.2  Results on Large-version Model

We compare the performance of DemoNSF with other baselines on the large-version model (i.e. T5-large and BART-large). Despite using a model with a larger parameter size, generative models still experience a significant decline in performance when faced with perturbed inputs, especially with fine-grained perturbations. As shown in Table 5, we can find that the model's performance declined by 37.42% for BART and 27.33% for T5 on Typos and 19.56% and 14.27% for BART and T5, respectively on Speech. Our approach also achieves impressive improvements on both fine-grained and coarse-grained perturbations. To be specific, DemoNSF introduces 5.18% F1 improvements on the average performance of all the perturbations input compared with InstructionNER based on T5 and 1.63% improvements compared with BARTNER based on BART.

### C.3  Details of Mixed Perturbations

For the mixed perturbations experiment on SNIPS, we also investigate the performance of DemoNSF on single perturbation (AppendIrr, Sub, and EntTypos). As shown in Table 6, we obtain similar conclusions. Specifically, our approach introduces significant improvements in fine-grained perturbations (e.g. 5.86% on EntTypos). While our approach also maintains exciting performance on coarse-grained perturbations (e.g. 2.51% improvements on AppendIrr).

| Methods | Clean | Sent App. | Word Sub. | Char Ent. | Char+Word Ent.+Sub. | Char+Sent Ent.+App. | Word+Sent App.+Sub. | Char+Word+Sent Ent.+Sub.+App. |
|---|---|---|---|---|---|---|---|---|
| BART | 79.43 | 65.95 | 71.20 | 57.84 | 58.00 | 47.27 | 50.28 | 38.36 |
| T5 | 94.12 | 83.97 | 85.13 | 65.90 | 57.44 | 56.79 | 73.47 | 47.93 |
| BARTNER | 86.34 | 69.33 | 77.22 | 62.28 | 54.83 | 49.10 | 58.92 | 42.25 |
| LightNER | 81.39 | 60.63 | 70.18 | 52.59 | 42.34 | 35.82 | 45.44 | 27.00 |
| InstructionNER | 94.69 | 84.32 | 84.78 | 66.93 | 57.87 | 58.89 | 74.45 | 50.75 |
| DemoNSF (BART) | 87.16($\pm$0.7) | 76.05($\pm$1.2) | 79.56($\pm$1.3) | 67.31($\pm$0.2) | 61.27($\pm$0.5) | 51.26($\pm$0.9) | 68.72($\pm$0.5) | 44.27($\pm$0.9) |
| DemoNSF (T5) | **94.75($\pm$1.7)** | **86.83($\pm$0.5)** | **86.81($\pm$0.2)** | **72.79($\pm$0.4)** | **63.59($\pm$0.3)** | **63.94($\pm$1.2)** | **77.69($\pm$0.7)** | **55.12($\pm$0.3)** |

Table 6: F1 scores with standard deviations under 3 kinds of single perturbations and 4 kinds of mixed perturbations on SNIPS.

## D Related Work

### D.1 Slot Filling

**Sequence Labeling Paradigm.** Initially, the slot filling task was commonly defined as a sequence labeling problem. Previous methods can be categorized into two types: one-stage and two-stage approaches. Specifically, one-stage approaches (Bapna et al., 2017; Shah et al., 2019; Lee and Jha, 2019) conduct slot filling individually for each slot type. It first generates word-level representations and the predictions are based on the concatenated features for each slot type. However, these methods only learn the surface mapping between entities and suffer from multi-prediction problems. To address these limitations, a branch of two-stage methods (Liu et al., 2020b; He et al., 2020b; Wang et al., 2021; Ma et al., 2022; Dong et al., 2023a; Wang et al., 2022a; Dong et al., 2023b) are proposed. Firstly, a coarse-grained binary sequence labeling model is used to identify all slot entities in the utterances. Subsequently, the entity value is mapped to the representation of the corresponding slot label in the semantic space in order to classify slot types effectively.

**Generative Framework.** Recently, some works (Wang et al., 2022b) have started to reformulating NER and slot filling tasks to sequence-to-sequence (seq2seq) tasks and integrate generative methods. BARTNER (He et al., 2020c) proposes a pointer-based seq2seq architecture, which converts the NER task to a unified sequence generation task and predicts entities from the input sentences and the corresponding type indexes LightNER(Chen et al., 2021) introduces prompt-tuning to the attention mechanism of BARTNER and achieves promising improvement in low-resource scenarios. Moreover, some prompt-based Generative methods (Lu et al., 2022; Xie et al., 2022b) have achieved strong performance in information extraction. Nevertheless, their exploration of generative frameworks on diverse input perturbations remains a blank area, hin-

dering their application in realistic task-oriented dialogue systems.

### D.2 Input Perturbation Problem

Recently, there has been a growing interest in enhancing the resilience of NLP systems to input perturbations. Moradi and Samwald (2021b) present empirical evaluations of the robustness of various NLP systems against input perturbations on synthetically generated benchmarks. Namysl et al. (2020, 2021) focus on the robustness of the NER model against Optical Character Recognition (OCR) noise and misspellings. Compared to other NLP systems, dialogue systems would face more diverse input noise due to more frequent interactions with users. Fang et al. (2020); Gopalakrishnan et al. (2020) investigate the robustness of dialogue systems on ASR noise, and Ruan et al. (2020); Li et al. (2020b); Huang and Chen (2020); Li et al. (2020a) mainly focus on the ASR-noise-robustness SLU models in dialogue systems.

Most previous studies (Moradi and Samwald, 2021a; Wu et al., 2021; Liu et al., 2020a) that investigated this robustness problem predominantly focused on rule-based synthetic datasets, which do possess certain limitations. Meanwhile, real-world dialogue systems encounter a wider range of perturbations due to frequent interactions with users. To further explore this direction, RADDLE (Peng et al., 2020) offers a crowd-sourced robustness evaluation benchmark for dialog systems, which includes various noisy utterances that existed in real dialogue scenarios. Liu et al. (2020a) introduced Language Understanding Augmentation, a methodology that incorporates four data augmentation techniques to simulate natural perturbations. To cope with more complex noisy scenarios, Dong et al. (2022a); Liu et al. (2023) investigate input perturbation problems on discriminative neural models. In this paper, we mainly focus on the performance of generative frameworks on input perturba-

tion problems.

### D.3 Demonstration-based learning

Demonstrations are first introduced by the GPT series (Radford et al., 2019; Brown et al., 2020), where a few examples are sampled from training data and transformed with templates into appropriately-filled prompts. Based on the task reformulation and whether the parameters are updated, the existing demonstration-based learning research can be broadly divided into three categories: In-context Learning (Brown et al., 2020; Zhao et al., 2021; Min et al., 2021; Wei et al., 2022), Prompt-based Fine-tuning (Liang et al., 2022; Dong et al., 2023c), Classifier-based Fine-tuning (Lee et al., 2021). However, these approaches mainly adopt demonstration-based learning in the fine-tuning that cannot make full use of the effect of demonstration-based learning.