# OpenReview forum: "DemoNSF: A Multi-task Demonstration-based Generative Framework for Noisy Slot Filling Task"
_EMNLP/2023/Conference — EMNLP 2023 Findings_

### Official Review · Reviewer_zUxk · 2023-08-03

**Soundness:** 3

**Excitement:**

3: Ambivalent: It has merits (e.g., it reports state-of-the-art results, the idea is nice), but there are key weaknesses (e.g., it describes incremental work), and it can significantly benefit from another round of revision. However, I won't object to accepting it if my co-reviewers champion it.

**Paper Topic And Main Contributions:**

This work proposes multi-task training to augment slot-filling training to capture input perturbations' features. It also constructs noisy demonstrations for the generative framework. The proposed approach performs better on diverse kinds of perturbations and various models (BART, T5, ChatGPT.)

**Reasons To Accept:**

1. The paper is well-written and easy to read.
2. The empirical study demonstrates the improvement of the proposed methods, especially on character-level and word-level perturbations.

**Reasons To Reject:**

1. The novelty seems a bit limited. If you introduce noises in pretaining and fine-tuning, it is expected to perform well on perturbed examples.
2. I personally think the ablation study/ChatGPT experiments/Ablation study are quite important, not sure why they appear in the appendix.

**Reproducibility:**

4: Could mostly reproduce the results, but there may be some variation because of sample variance or minor variations in their interpretation of the protocol or method.

**Reviewer Confidence:**

3: Pretty sure, but there's a chance I missed something. Although I have a good feel for this area in general, I did not carefully check the paper's details, e.g., the math, experimental design, or novelty.

---

> ### Author Rebuttal · Authors · 2023-08-27
>
> Thank you very much for your valuable suggestions and questions.
> **Regarding your first question**:
>  "The novelty seems a bit limited. If you introduce noises in pretaining and fine-tuning, it is expected to perform well on perturbed examples."
> **Our response**:
> We greatly appreciate the question you've raised. In order to examine whether the augmented data in pre-training and fine-tuning stages significantly enhances the model's robustness, we conduct an experiment where augmented data was mixed with the training data for joint training. The experimental results are presented in the table below:
>
> | Methods | Clean | Verbose | Paraphrase | Simplification | Typos | Speech | Perturbed-Avg. |
> | :---- | :----: | :----: | :----: | :----: | :----: | :----: | :----: |
> | T5 (baseline) | 95.49 | 81.34 | 89.13 | 83.73 | 62.43 | 81.13 | 79.55 |
> | DemoNSF (T5) | 95.72 |  82.37 | 89.98 | 89.49 | 76.63 | 87.55 | 85.20 |
> | Data Aug. | 95.11 | 81.49 | 89.62 | 85.38 | 65.13 | 82.88 | 80.90 (-4.3) |
>
> The results indicate that solely performing data augmentation does not lead to a substantial improvement in the model's robustness against perturbed data, instead, it compromises the performance on clean data. We believe that rule-based data augmentation (such as random character augmentation and random deletion) deviates significantly from real-world test data, causing the model trained on mixed data to learn only superficial predictive patterns from the augmented data, without truly grasping the essence of perturbations. Consequently, when faced with real-world data, the presence of a gap prevents significant performance improvement.
> In contrast, our proposed approach is capable of learning the semantic structural information and inherent forms of perturbed data at three different levels of granularity (character, word, sentence). This enables the model to better capture intrinsic details such as perturbation granularity and patterns when dealing with real-world perturbed data, resulting in enhanced predictive outcomes.
>
> **Regarding your second question**:
> "I personally think the ablation study/ChatGPT experiments/Ablation study are quite important, not sure why they appear in the appendix."
> **Our response**:
> Due to the limitations in the length of the paper, we have placed the primary analytical experiments in the main body of the text, while the extensible experiments (such as the ChatGPT experiment and ablation study) have been included in the appendix. We sincerely appreciate your valuable suggestion, and we will optimize the structure of the paper based on your recommendation.
> Once again, we express our gratitude for your two suggestions. We are committed to further enhancing the structure and content of this paper, incorporating additional experiments, and striving to elevate the quality of our research.

---

### Official Review · Reviewer_vQ9E · 2023-08-04

**Soundness:** 3

**Excitement:**

3: Ambivalent: It has merits (e.g., it reports state-of-the-art results, the idea is nice), but there are key weaknesses (e.g., it describes incremental work), and it can significantly benefit from another round of revision. However, I won't object to accepting it if my co-reviewers champion it.

**Paper Topic And Main Contributions:**

This paper proposes a framework called DemoNSF to address the issue of input perturbations in slot filling tasks in dialogue systems.

The existing models are trained on perturbation-free datasets, which leads to decreased performance when faced with different human expressions and input perturbations.

To tackle this limitation, the authors propose the DemoNSF framework, which includes three noisy auxiliary tasks: noisy recovery, random mask, and hybrid discrimination. The framework formulates the slot filling task as a sequence-to-sequence generation guided by noisy task demonstrations. It selects semantically similar examples for each query, converts them into natural demonstration sentences, and encodes them along with the input text to learn the semantic structure of perturbations.

Experimental results demonstrate that DemoNSF outperforms baseline methods, achieves strong generalization, and effectively addresses fine-grained and coarse-grained perturbations, as well as mixed perturbation scenarios.

**Reasons To Accept:**

The three main contributions of this paper can be summarized as follows:
	• Investigating the Effects of Input Perturbations: This paper is the first to comprehensively investigate the effects of diverse input perturbations on generative frameworks in slot filling tasks. It highlights the vulnerability of existing prompt-based generative methods when faced with different human expressions.
	• Multi-task Demonstration-based Generative Framework: The authors propose a simple yet unified multi-task demonstration-based generative framework called DemoNSF. It includes three novel noisy auxiliary tasks and a noisy demonstration construction strategy, which enhance the model's robustness and adaptability to perturbed inputs in real-world dialogue scenarios.
	• Improved Performance and Generalization: DemoNSF outperforms all baseline methods and demonstrates strong generalization. The extensive analysis provided in the paper also offers empirical guidance for the practical application of generative frameworks.

**Reasons To Reject:**

The experiment is not enough. The authors build T5 and BART as baseline models, however, The GPT-style models are not considered in this paper. Maybe it is interesting to include some GPT-style models.


**Reproducibility:**

3: Could reproduce the results with some difficulty. The settings of parameters are underspecified or subjectively determined; the training/evaluation data are not widely available.

**Reviewer Confidence:**

3: Pretty sure, but there's a chance I missed something. Although I have a good feel for this area in general, I did not carefully check the paper's details, e.g., the math, experimental design, or novelty.

---

> ### Author Rebuttal · Authors · 2023-08-27
>
> Thank you greatly for your valuable suggestions and your insights are truly appreciated. To validate the scalability and adaptability of our proposed method, we have conducted additional experiments using GPT-2 small. The outcomes of these experiments are presented in the subsequent table:
>
> | Methods | Clean | Verbose | Paraphrase | Simplification | Typos | Speech | Perturbed-Avg. |
> | :---- | :----: | :----: | :----: | :----: | :----: | :----: | :----: |
> | BART | 95.28 | 77.87 | 82.72 | 82.95 | 53.90 | 73.57 | 74.20 |
> | T5 | 95.49 | 81.34 | 89.13 | 83.73 | 62.43 | 81.13 | 79.55 |
> | GPT-2 | 95.37 | 80.52 | 85.66 | 82.98 | 60.19 | 77.78 | 77.43 |
> | DemoNSF(BART) | 95.71 | 78.83 | 88.29 | 86.01 | 65.60 | 82.48 | 80.24 |
> | DemoNSF(T5) | 95.72 |  82.37 | 89.98 | 89.49 | 76.63 | 87.55 | 85.20 |
> | DemoNSF(GPT-2) | 95.66 | 81.95 | 87.63 | 87.02 | 69.75 | 86.31 | 82.53 |
>
> The experimental results demonstrate that our approach achieves state-of-the-art results on GPT-style models as well. This also indicates the plug-and-play, scalable nature of our method within the generative framework.
> Once again, we express our gratitude for your valuable suggestions. We will continue to refine our article's content and strive to enhance the quality and comprehensiveness of our research and experiments.

---

### Official Review · Reviewer_8JBt · 2023-08-04

**Soundness:** 4

**Excitement:**

3: Ambivalent: It has merits (e.g., it reports state-of-the-art results, the idea is nice), but there are key weaknesses (e.g., it describes incremental work), and it can significantly benefit from another round of revision. However, I won't object to accepting it if my co-reviewers champion it.

**Paper Topic And Main Contributions:**

This paper presents DemoNSF, a multi-task demonstration-based generative framework for the noisy slot-filling task in practical dialogue scenarios. It proposes three novel noisy auxiliary tasks in the pre-training stage to capture semantic structural information of input perturbations at different granularities. Moreover, it conducts extensive experiments on two benchmarks and demonstrates that DemoNSF outperforms several baseline methods by a large margin.

**Reasons To Accept:**

1. The paper introduces an effective multi-task demonstration-based generative framework for the noisy slot-filling task in practical dialogue scenarios.
2. The paper is well-written and easy to follow. The authors provide clear explanations of the proposed framework and the experiments conducted.

**Reasons To Reject:**

No particular reason to reject.

**Reproducibility:**

4: Could mostly reproduce the results, but there may be some variation because of sample variance or minor variations in their interpretation of the protocol or method.

**Reviewer Confidence:**

3: Pretty sure, but there's a chance I missed something. Although I have a good feel for this area in general, I did not carefully check the paper's details, e.g., the math, experimental design, or novelty.

---

> ### Author Rebuttal · Authors · 2023-08-27
>
> We sincerely appreciate your valuable appraisal and acknowledgment of our work. We are committed to enhancing and enriching our study with additional experiments and analyses, in accordance with the recommendations from all the reviewers, to further substantiate the robustness of our findings.

---

### Meta-Review · Area_Chair_nx9x · 2023-09-18

**Recommendation:** 3

**Metareview:**

Summary Evaluation:

The paper introduces DemoNSF, a multi-task demonstration-based generative framework for the noisy slot-filling task in practical dialogue scenarios. It proposes three novel noisy auxiliary tasks in the pre-training stage to capture semantic structural information of input perturbations at different granularities. Experimental results demonstrate that DemoNSF outperforms baseline methods and achieves strong generalization.

Pros:
- Addresses an important issue of input perturbations in slot filling tasks in dialogue systems.
- Proposes a novel multi-task demonstration-based generative framework called DemoNSF that improves performance and generalization.
- Clear and well-written paper with thorough experiments conducted.
- First comprehensive investigation of the effects of diverse input perturbations on generative frameworks in slot filling tasks.

Cons:
- The novelty seems to be limited, and the improvements could be expected.
- While it uses T5 and BART as baseline models, GPT-style models are not considered.
- Some important experimental results such as ablation studies and ChatGPT experiments are placed in the appendix instead of the main paper.

After considering the updated scores of the reviewers for soundness and excitement, this paper presents a valuable contribution to the field of NLP and dialogue systems. The proposed framework, DemoNSF, shows improved performance and generalization over existing models in addressing input perturbations in slot filling tasks. However, the authors could consider incorporating GPT-style models in their comparison and moving some experimental results from the appendix to the main paper to strengthen the work further.

---

### Decision · Program_Chairs · 2023-10-07

**Decision:**

Accept-Findings

**Comment:**

Summary Evaluation:

The paper introduces DemoNSF, a multi-task demonstration-based generative framework for the noisy slot-filling task in practical dialogue scenarios. It proposes three novel noisy auxiliary tasks in the pre-training stage to capture semantic structural information of input perturbations at different granularities. Experimental results demonstrate that DemoNSF outperforms baseline methods and achieves strong generalization.

Pros:
- Addresses an important issue of input perturbations in slot filling tasks in dialogue systems.
- Proposes a novel multi-task demonstration-based generative framework called DemoNSF that improves performance and generalization.
- Clear and well-written paper with thorough experiments conducted.
- First comprehensive investigation of the effects of diverse input perturbations on generative frameworks in slot filling tasks.

Cons:
- The novelty seems to be limited, and the improvements could be expected.
- While it uses T5 and BART as baseline models, GPT-style models are not considered.
- Some important experimental results such as ablation studies and ChatGPT experiments are placed in the appendix instead of the main paper.

After considering the updated scores of the reviewers for soundness and excitement, this paper presents a valuable contribution to the field of NLP and dialogue systems. The proposed framework, DemoNSF, shows improved performance and generalization over existing models in addressing input perturbations in slot filling tasks. However, the authors could consider incorporating GPT-style models in their comparison and moving some experimental results from the appendix to the main paper to strengthen the work further.